# SPARSE-ART: ENABLING INTERACTABLE ARTICULATED OBJECTS FROM UNPOSED SPARSE-VIEW INPUT

## ABSTRACT

Articulated object perception is essential for intelligent agents in robotics, embodied AI, and augmented reality, yet reconstructing their geometry and kinematics from sparse RGB images remains a significant challenge. Traditional optimization-based methods, such as those using NeRFs or 3DGS, deliver high fidelity but demand time-intensive per-object optimization, while data-driven approaches suffer from limited 3D datasets, restricting generalization to real-world scenarios. To address these limitations, we introduce a novel, fully training-free and feed-forward framework that reconstructs and analyzes articulated objects from 1-4 sparse, unposed RGB images per state, captured in two states of the object. Our approach leverages foundation models for unified geometric-semantic processing without any fine-tuning, enabling efficient inference for part correspondences and joint classification, followed by lightweight optimization for parameter estimation. Dataset-independent with a fully training-free and feed-forward design that eliminates the need for per-object training or extensive iterations, our method effectively bridges synthetic-to-real gaps, achieving superior performance on real-world objects. By integrating end-to-end zero-shot reconstruction with advanced inference and optimization, it provides an efficient, robust solution for articulation modeling, advancing scalable applications in robotics.

## 1 INTRODUCTION

The ability to perceive and model articulated objects is fundamental for intelligent agents to interact with the world (Liu et al., 2025c; Weng et al., 2024; Liu et al., 2025b). High-quality, large-scale acquisition of articulated object models is a critical enabler for a myriad of applications, from robot interaction and manipulation in embodied AI to realistic asset creation for augmented reality and gaming. However, efficiently capturing the geometry and kinematic structures of such objects remains a significant challenge in computer vision and robotics.

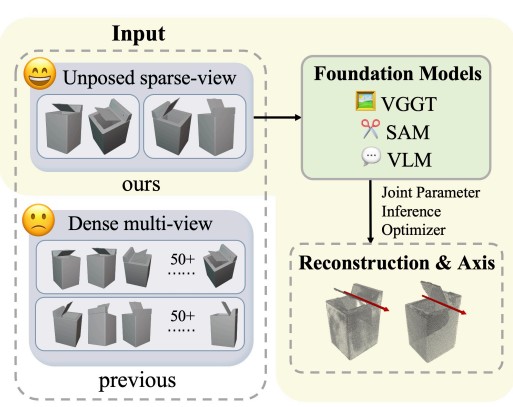

Figure 1: **Sparse-Art** introduces a novel setting for articulated object reconstruction, achieving zero-shot, feed-forward inference from sparse, unposed RGB images via foundation models, without requiring dense inputs.

Existing approaches to this problem can be broadly categorized into two main directions. The first encompasses optimization-based methods that reconstruct articulated objects by fitting models to observations. These techniques often leverage implicit representations like Neural Radiance Fields (NeRFs) (Liu et al., 2023; Lewis et al., 2024) or explicit ones like 3D Gaussian Splatting (3DGS) (Liu et al., 2025d; Lin et al., 2025). While capable of high fidelity, their primary drawback is computational cost; they typically require a time-consuming, per-object optimization process on the order of ten minutes, rendering them impractical for large-scale applications. The second direction relies on large-scale data and foundation models to learn gen-

eralizable priors for articulation estimation (Qiu et al., 2025; Yu et al., 2025; Patil et al., 2023). A major bottleneck for these methods is the scarcity of large, diverse, and high-quality 3D datasets of articulated objects, which limits their generalization to novel object categories and real-world scenarios.

To overcome these limitations, we propose a new paradigm that circumvents the need for both lengthy optimization and extensive training data. Our approach is made possible by the confluence of recent breakthroughs in foundation models. First, the advent of powerful, generalizable sparse-view 3D reconstruction models, such as Dust3R (Wang et al., 2024c) and VGGT (Maggio et al., 2025), has dramatically simplified the process of obtaining a 3D point cloud from just a handful of images. Second, the rapid advancement of Vision-Language Models (VLMs) (Radford et al., 2021; Alayrac et al., 2022; Li et al., 2023a) has provided an unprecedented source of visual and semantic knowledge that can be leveraged to understand object composition and functionality. We posit that these powerful foundation models make a fully training-free pipeline for articulated object reconstruction not only possible, but also highly effective. Our method takes sparse, unposed RGB images of an object in two states and first performs a unified geometric-semantic lifting, using VGGT and SAM (Kirillov et al., 2023) to generate semantically-aware 3D point clouds. Next, a VLM-powered inference module reasons over these representations to perform zero-shot part correspondence and joint type classification. Finally, these semantic predictions guide a lightweight, classification-aware optimizer to efficiently infer the precise articulation parameters, making a fully training-free reconstruction possible.

Our proposed framework offers several distinct advantages. It is entirely training-free, eliminating the dependency on curated datasets. It is also fast, completing the reconstruction and articulation analysis of an object in just a few minutes—much faster than optimization-based techniques, which often require 10 minutes or more. Furthermore, our method demonstrates exceptional robustness in bridging the synthetic-to-real domain gap. While many prior works excel on synthetic data but suffer a significant performance drop on real-world captures, our approach maintains consistent and superior performance on real-world data. Our main contributions can be summarized as follows:

- We introduce a novel, end-to-end framework for articulated object reconstruction and analysis from sparse RGB images that is entirely zero-shot and training-free.

- We propose a unified geometric-semantic lifting mechanism that seamlessly injects 2D part-level semantics into a 3D point cloud reconstruction in a single forward pass, leveraging foundation models like VGGT and SAM to create an articulation-aware representation without any fine-tuning.

- We design a zero-shot semantic inference module that employs VLMs for high-level reasoning to determine part correspondence and joint types, coupled with an efficient, classification-guided optimizer that enables fast and robust parameter estimation.

## 2 RELATED WORK

### 2.1 VISION LANGUAGE MODELS

Vision-Language Models (VLMs) have evolved from foundational models like CLIP, enabling zero-shot classification via contrastive learning (Radford et al., 2021), to advanced systems with few-shot learning, instruction tuning, and multimodal capabilities, such as Flamingo, BLIP-2, LLaVA, InternVL-3, Qwen2-VL, and Elysium (Alayrac et al., 2022; Li et al., 2023b;a; Zhu et al., 2025; Bai et al., 2025; Wang et al., 2024a).

These advancements support diverse object-level perception tasks. In open-vocabulary object detection, Grounding DINO and YOLO-World achieve state-of-the-art results on open-set benchmarks with real-time performance (Liu et al., 2024; Cheng et al., 2024). Visual grounding and referring expression comprehension enable precise localization from text, as in Qwen2.5-VL for coordinate-based pinpointing and Elysium for video extensions like Referring Single Object Tracking (Bai et al., 2025; Wang et al., 2024a; Liao et al., 2025). At the pixel level, referring expression segmentation is advanced in SegVLM (via CogVLM) and LGRS (using specialized tokens for language-mask linking) (Wang et al., 2024d; Yao et al., 2025a).

## 2.2 SPARSE VIEW 3D RECONSTRUCTION

The challenge of reconstructing 3D scenes from sparse or unconstrained images has shifted from traditional iterative optimization to generalizable, feed-forward models (Younis & Cheng, 2025). Early methods used multi-stage pipelines requiring pre-calibrated cameras, but Neural Radiance Fields (NeRFs) enabled breakthrough novel view synthesis, though limited by real-time performance and sparse inputs (Mildenhall et al., 2021; Liu et al., 2025a). This prompted a transition to explicit representations like 3D Gaussian Splatting (3DGS) (Kerbl et al., 2023). The evolution started with pointmap-based approaches, such as the DUSt3R series, which introduced pose-free, unconstrained reconstruction via end-to-end networks generating pointmaps from arbitrary images (Wang et al., 2024c). Subsequent works advanced to versatile 3DGS primitives, including Splatt3r and NoPoSplat (Smart et al., 2024; Ye et al., 2024). Splatt3r extended DUSt3R to infer all Gaussian attributes, using loss masking for uncalibrated, in-the-wild images (Smart et al., 2024), while NoPoSplat simplified it with pure photometric loss and canonical space, achieving superior quality without ground-truth depth or pose supervision (Ye et al., 2024). Culminating this trajectory, the Visual Geometry Grounded Transformer (VGGT) unifies direct inference of full 3D attributes in one pass, often surpassing optimization-based methods. Extensions include VGGT-SLAM for dense RGB SLAM (Maggio et al., 2025) and StreamVGGT for real-time 4D video reconstruction (Zhuo et al., 2025). Related 4D works leverage transformers for interactive hands and humans (Lin et al., 2024; Goel et al., 2023), alongside motion-aware enhancements for dynamic 3DGS (Guo et al., 2024; Lee et al., 2024).

## 2.3 ARTICULATION PARAMETER ESTIMATION

Articulated objects, featuring multiple rigid parts connected by joints, pose greater 3D reconstruction challenges than rigid-body pose estimation (**?**Liu et al., 2025b; Jiang et al., 2022). Early methods relied on CAD models or category-specific datasets, limiting generalization (Nie et al., 2022; **?**). Recent approaches emphasize generalizable, self-supervised, and open-vocabulary methods with novel representations (Le et al., 2024; Qiu et al., 2025).

Implicit representations advance the field: NeRF-based techniques (e.g., PARIS, NARF24) infer joints via shared models and segmentation but require costly multi-view setups (Liu et al., 2023; Lewis et al., 2024). Efficient 3DGS methods like ArtGS, SPLART, and extensions (e.g., ReArtGS, RigGS, IAAO) enable self-supervised segmentation, kinematics, and multi-part modeling, yet struggle with multi-state integration and accuracy (Liu et al., 2025d; Lin et al., 2025; Wu et al., 2025; Wang et al., 2025a; Yao et al., 2025b; Zhang & Lee, 2025).

Explicit and interactive methods improve robustness: PartRM, RPMArt, GAMMA, and Structure from Action handle dynamics, noise, and manipulations, though vulnerable to depth issues and sim-reality gaps (Gao et al., 2025; Wang et al., 2024b; Yu et al., 2024; Nie et al., 2022).

Large foundation models enable open-vocabulary articulation: Articulate AnyMesh, GAPS, and RoSI adapt meshes, refine segmentation, and infer interiors, despite generalization and computational limitations (Qiu et al., 2025; Yu et al., 2025; Patil et al., 2023).

## 3 METHOD

To address the core challenges of sparse input data and computational constraints in embodied intelligence, we propose a novel end-to-end framework that utilizes sparse, unposed RGB images (typically 1–4 per state) of articulated objects captured at two distinct joint states. This allows the construction of semantically enriched 3D point clouds, denoted as $P_s$ and $P_e$, where the subscript 's' represents the start state and 'e' signifies the end state, respectively. From these representations, we infer a set of articulation parameters $\Psi$, which includes joint type, axis, and motion. Our primary contribution is a fully zero-shot, training-free paradigm that seamlessly integrates foundation models without fine-tuning, effectively merging 2D motion semantics into 3D representations to support robust, semantics-guided articulation inference in previously unseen scenes. This method bridges the gap between conventional decoupled approaches and more efficient embodied perception, as demonstrated in Figure 2. The subsequent sections elaborate on our methodological innovations.

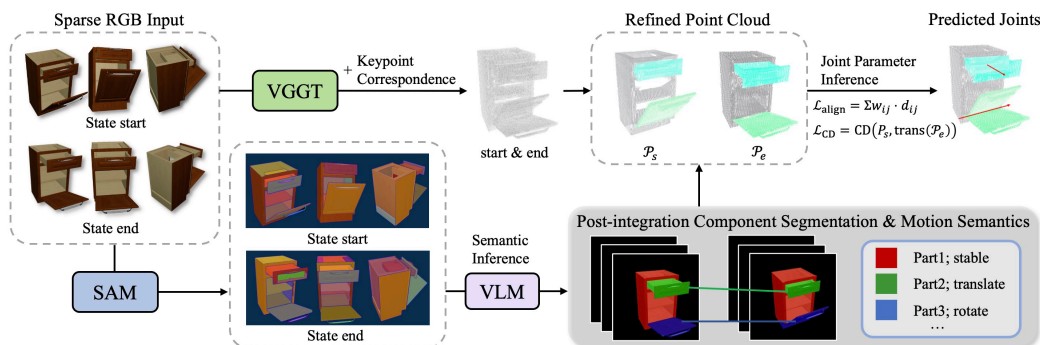

Figure 2: **The overview of Sparse-Art.** Our training-free framework reconstructs articulated objects from sparse RGB images in two states by performing unified geometric-semantic lifting with VGGT and SAM to create semantically-aware 3D point clouds, followed by VLM-powered zero-shot inference for part correspondence and joint classification, and a lightweight optimizer for precise articulation parameter estimation.

## 3.1 UNIFIED GEOMETRIC-SEMANTIC LIFTING FOR ARTICULATION-AWARE REPRESENTATIONS

Conventional approaches decouple 2D segmentation from 3D reconstruction, resulting in semantic inconsistencies and degraded performance under sparse views. In contrast, we propose a unified lifting mechanism as the cornerstone of our framework, which seamlessly transfers and fuses semantic cues from 2D images into 3D point clouds in a zero-shot manner, fostering articulation-aware representations that support all downstream inference without requiring any training or per-scene adaptation. To achieve this, we harness the attention-based capabilities of VGGT (Wang et al., 2025b) to generate initial 3D reconstructions $\mathcal{M}_s$ and $\mathcal{M}_e$ from sparse RGB inputs, while injecting part-level semantics from the Segment Anything Model (SAM) (Kirillov et al., 2023) in a prompt-free manner. The semantic lifting from 2D masks to 3D point clouds leverages the inherent pointmap structure of VGGT, which establishes a bijective correspondence between the rasterized 2D pixel grid and the flattened 1D ordering of the reconstructed point cloud. Formally, given a 2D semantic mask $\mathbf{M} \in \mathcal{S}^{H \times W}$ (where $\mathcal{S}$ denotes the semantic label space) and the VGGT-generated point cloud $\mathcal{P} \in \mathbb{R}^{HW \times C}$ (with $C$ channels, e.g., 3 for XYZ coordinates), the enriched point cloud $\mathcal{P}^{sem} \in \mathbb{R}^{HW \times (C+1)}$ (appending a semantic channel) is obtained via the flattening isomorphism:

$$\mathcal{P}^{sem}_k = \left( \mathcal{P}_k, \mathbf{M}_{\lfloor k/W \rfloor, \, k \bmod W} \right), \quad \forall k \in \{0, 1, \ldots, HW - 1\}, \tag{1}$$

where the floor division and modulo operations encode the implicit row-major raster order mapping from 2D pixel indices $(i, j)$ to the 1D point index $k = i \cdot W + j$. This zero-shot transfer preserves spatial alignment without explicit geometric inversion or trainable parameters.

Furthermore, to ensure cross-state coordinate consistency—a common failure point in sparse reconstructions—we incorporate a dynamic tracking module that aligns $\mathcal{P}_s$ and $\mathcal{P}_e$ using keypoint correspondences from LightGlue (Lindenberger et al., 2023). This alignment establishes a canonical frame, providing a robust foundation for articulation parameter estimation without iterative refinements or training dependencies. Overall, this lifting module not only bridges 2D-to-3D semantics but also enables seamless integration with high-level reasoning, forming a cohesive basis for zero-shot articulation understanding.

## 3.2 ZERO-SHOT SEMANTIC INFERENCE VIA VISION-LANGUAGE INTEGRATION

Building directly on the articulation-aware 3D representations from Section 3.1, we contribute a zero-shot semantic inference module powered by vision-language models (VLMs), which extracts high-level articulation knowledge—such as part counts, correspondences, and joint types—without any supervision, training, or optimization-heavy processes. This fusion of semantic reasoning with

the pre-lifted geometric structure represents a key advancement for handling unseen objects in embodied intelligence, eliminating the need for per-scene adaptation.

**Part Count Estimation.** Traditional methods assume fixed joint typologies, limiting adaptability to novel scenarios. We address this by employing VLM-based analysis to robustly estimate the number of rigid parts $n_{\text{parts}}$ across states in a zero-shot manner. By querying sampled image pairs with targeted prompts (e.g., "Count the number of movable components") and aggregating responses via mode selection, our approach mitigates VLM inconsistencies, offering a flexible foundation for part-level reasoning without training.

**Prompt-Guided Part Segmentation and Correspondence.** Building on this, we develop a structured prompting strategy for VLMs to perform part segmentation and cross-state correspondence, overcoming occlusion and sparsity challenges without manual intervention or supervision. The process begins with holistic structural analysis of multi-view images across states, followed by mask-wise labelling of SAM outputs to resolve fragmentation and ensure spatial consistency. As detailed in Table 1, parts are assigned identifiers $p_k$ where static elements are labelled 1 and dynamic ones 2 to $n$. Unlike prior methods (Liu et al., 2023; Weng et al., 2024; Liu et al., 2025d) that assume isolated objects or require manual pre-separation of scenes, our 2D-to-3D semantic transfer enables robust handling of real-world data with complex backgrounds, such as unprocessed real-scan inputs. This prompt-based paradigm eliminates annotation dependencies, enabling automated, coherent semantic mapping that outperforms conventional clustering techniques in a training-free setting.

Table 1: Prompt Design for Part Correspondence Establishment

| Component | Specification |
|---|---|
| Input Configuration | Multi-view images from $N$ distinct viewpoints $S = \{s_1, s_2, \ldots, s_N\}$, covering both initial and terminal states (yielding $2N$ images in total) |
| Structural Analysis | Comprehensive evaluation of articulated object mechanics and kinematic behavior of components across states |
| Mask Processing | For the $i$-th image ($i = 1, \ldots, 2N$), each of the j masks $M_{ij}$ is assigned a part label $p_k$, where $k = 0, \ldots, n$, where $n$ denotes the total number of parts ($n_{parts}$) as introduced in part count estimation |
| Correspondence Criteria | Emphasis on spatial consistency: identical parts retain persistent labels throughout state transitions |
| Output Format | Structured classification: each mask is assigned a part identifier $p_k \in \mathbb{Z}_{\geq 0}$ defined as: $p_k = \begin{cases} 0, & \text{not part of the object} \\ 1, & \text{static part} \\ 2, \ldots, n, & \text{dynamic parts} \end{cases}$ |
| Execution Protocol | Automated mask-wise processing integrated with cross-state validation and fragmentation handling |

**Joint Type Classification** Finally, the VLM infers joint types (revolute or prismatic) and dynamic/static statuses for each part $\mathcal{P}i$ using specialized prompts (e.g., "Identify articulation semantics"). The resulting semantics $T_1, \ldots, T n_{\text{parts}}$ guide parameter prediction, ensuring our framework's zero-shot adaptability to diverse articulated structures without any training requirements.

### 3.3 EFFICIENT ARTICULATION PARAMETER INFERENCE AND REFINEMENT

Departing from computationally intensive iterative optimizations like ICP, we propose a classification-guided, feed-forward inference strategy that refines VLM-derived initial estimates using a lightweight optimizer, achieving precise axis and motion predictions with enhanced robustness—particularly for revolute joints—all without the need for training or fine-tuning.

We first refine point correspondences in $\mathcal{P}$ through outlier removal and clustering, yielding $\mathcal{P}_{\text{refined}}$. Then, leveraging joint types $T_i$ from Section 3.2, we apply type-specific transformations (rotational

for revolute, translational for prismatic) and optimize via the Hungarian algorithm. A novel contribution is our distance-proportional weighting in the loss for revolute joints, which prioritizes reliable long-distance pairs to mitigate noise-induced ambiguities, defined as:

$$w_{ij} = \frac{N \cdot \left( \frac{d_{ij} - d_{\min}}{d_{\max} - d_{\min}} + \beta \right)}{\sum_{k=1}^{N} \left( \frac{d_k - d_{\min}}{d_{\max} - d_{\min}} + \beta \right)}, \quad \beta = 0.1. \tag{2}$$

For prismatic joints, uniform weighting suffices. The total loss balances alignment $\mathcal{L}_{\text{align}} = \sum w_{ij} \cdot d_{ij}$ with Chamfer Distance $\mathcal{L}_{\text{CD}}$:

$$\mathcal{L} = \mathcal{L}_{\text{align}} + \alpha \cdot \mathcal{L}_{\text{CD}}. \tag{3}$$

This optimizer, initialized from foundation model outputs, enables end-to-end processing via re-projection errors and efficient inference, underscoring our method's suitability for resource-constrained embodied applications in a fully training-free manner.

## 4 EXPERIMENTS

**Datasets.** We evaluate our method on three datasets: (1) PARIS (Liu et al., 2023), a two-part dataset featuring articulated objects with one static and one movable part, including 10 synthetic objects from the PartNet-Mobility dataset (Xiang et al., 2020) and 2 real-world objects captured using the MultiScan toolset (Mao et al., 2022); (2) DTA-Multi (Weng et al., 2024), which contains 2 synthetic multi-part articulated objects from PartNet-Mobility, each with one static and two movable parts; (3) ArtGS-Multi (Liu et al., 2025d), featuring 5 complex articulated objects from PartNet-Mobility with three to six movable parts.

**Metrics.** To evaluate the joint axis, we compute the angular error (Axis Ang) in degrees for both joint types, which measures the orientation discrepancy between the predicted and ground-truth axis directions, ranging from 0° to 90°. For revolute joints, we additionally report the position error (Axis Pos) in centimeters, defined as the minimum distance between the predicted and reference rotation axes, accounting for the pivot point location.

**Baselines.** We compare our method with recent approaches. The single-joint baselines, which are capable of processing only one articulated part at a time, comprise Ditto (Jiang et al., 2022) and PARIS (Liu et al., 2023). Each of these pipelines was executed individually for each joint: in every run, the input images depict motion of a single joint while keeping other parts static, and the per-joint outputs were subsequently merged for assessment. For multi-joint scenarios, ArtGS (Liu et al., 2025d) serves as a baseline that can handle multiple movable parts in one pass. However, since ArtGS operates on multi-input data by default, whereas our novel framework targets sparse output, we adapted ArtGS by reducing its input image count to align with this constraint. This modified version is referred to as ArtGS* in our evaluations.

**Implementation Details.** For the baseline methods, Ditto and PARIS, we used the full set of input images and ran their experiments with the default parameters as specified by their official implementations. This ensures a fair and direct comparison with their published results. For our method, we randomly selected three images per state for each task. We specifically ensured that the poses of one set of images were close to each other (the camera did not rotate more than a quarter turn around the object) to facilitate the alignment between $\mathcal{P}_s$ and $\mathcal{P}_e$. All quantitative results presented in our tables were obtained using a single NVIDIA RTX-3090 GPU for training and inference. We also verified that our model can be run on an NVIDIA RTX-4060 GPU with a comparable execution time, demonstrating its compatibility with a wider range of hardware. During the Visual-Language Model (VLM) processing stage, we utilized the Gemini 2.5 Pro to handle the multimodal inputs.

### 4.1 SAPIEN OBJECTS RESULTS

**Single-Articulation Objects.** Our approach is tailored for sparse-view reconstruction of articulated objects—a challenging setting in which conventional multi-view methods often fail due to

Table 2: **Quantitative evaluation on PARIS.** For fair and consistent comparison, results for Ditto (Jiang et al., 2022) and PARIS (Liu et al., 2023) are sourced from (Liu et al., 2025d), where all methods are evaluated under identical conditions. Following prior protocols, metrics are averaged over 10 trials at highly visible joint configurations. To ensure equity, PARIS incorporates depth input as per its original paper. Axis Position Error is omitted for prismatic joints (e.g., Blade and Storage), as it does not apply to translational motion. Lower values (↓) indicate better performance; best results are in **bold**.

| Metric | Method | Sapien Objects | | | | | | | | | | |
| --- | --- | --- | --- | --- | --- | --- | --- | --- | --- | --- | --- | --- |
| | | FoldChair | Fridge | Laptop | Oven | Scissor | Stapler | USB | Washer | Blade | Storage | All |
| | Ditto | 89.35 | 89.30 | 3.12 | 0.96 | 4.50 | 89.86 | 89.77 | 89.51 | 79.54 | 6.32 | 54.22 |
| Axis Ang | PARIS | 15.79 | **2.93** | **0.03** | 7.43 | 16.62 | 8.17 | **0.71** | 18.40 | 41.28 | **0.03** | 11.14 |
| | Ours | **1.90** | 7.08 | 1.15 | **0.59** | **4.32** | **2.80** | 2.89 | **1.47** | **2.21** | 1.65 | **2.61** |
| | Ditto | 3.77 | 1.02 | 0.01 | 0.13 | 5.70 | **0.20** | 5.41 | 0.66 | - | - | 2.11 |
| Axis Pos | PARIS | 0.25 | 1.13 | **0.00** | **0.0** | 1.59 | 4.67 | 3.35 | 3.28 | - | - | 1.79 |
| | Ours | **0.06** | **0.39** | 0.13 | 0.26 | **0.53** | 0.60 | **0.91** | **0.17** | - | - | **0.38** |

insufficient visual information. As summarized in Table 2 and Figure 3, our method achieves competitive performance while exhibiting an advantage in computational efficiency. It successfully infers both articulation structure and object state from very few input images within a short processing time. These results underscore the applicability of our method in resource-limited settings, such as in robotic systems and low-budget data collection environments. Although our method does not outperform some recent state-of-the-art techniques that are optimized for dense-view inputs—particularly on synthetic datasets with limited texture variation—it establishes a new state of the art for reconstruction from extremely sparse image sets.

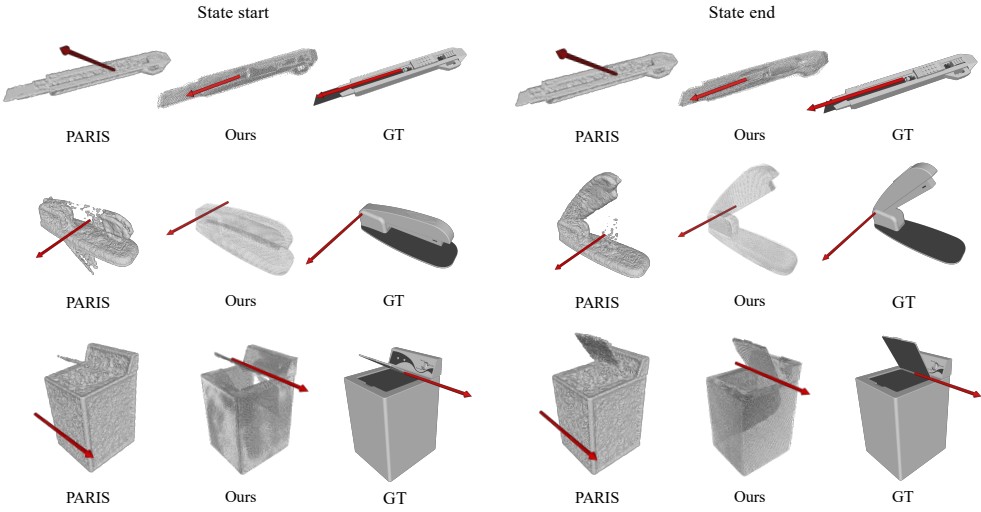

Figure 3: **Qualitative visualizations of PARIS objects.** We present reconstruction comparisons between PARIS and our Sparse-Art on synthetic objects from the PARIS dataset: Blade (Top), Stapler (Middle), and Washer (Bottom). Even with multi-view inputs, PARIS struggles with accurate mesh reconstruction in low-visibility or challenging states. In contrast, our sparse-view method leverages unified geometric-semantic lifting and cross-state alignment to enhance reconstruction quality for both states, achieving more robust and detailed results despite limited inputs.

**Multi-Articulation Objects.** To further evaluate the robustness of our method, we performed extended experiments on two datasets featuring multi-articulation objects. The results, presented in Tables 3 and Figure 4, demonstrate our model's ability to handle complex scenarios involving multiple joints and interconnected components. Both qualitative and quantitative evaluations confirm

Table 3: **Quantitative evaluation on ArtGS-Multi and DTA-Multi.** To ensure a fair and consistent comparison, all results are sourced from (Liu et al., 2025d), where the data are evaluated under identical experimental conditions. The metrics are computed as the mean over the available data points, following the averaging procedure described in the source. For each dataset, which comprises multiple joint axes, the evaluation aligns with established protocols.

| Metric | Method | ArtGS-Multi | | | | | DTA-Multi | | |
| | | Oven 101908 (4 parts) | Storage 45503 (4 parts) | Storage 47648 (7 parts) | Table 25493 (4 parts) | Table 31249 (5 parts) | Fridge 10489 (3 parts) | Storage 47254 (3 parts) | All |
|---|---|---|---|---|---|---|---|---|---|
| | DTA | 17.83 | 51.18 | 19.07 | 24.35 | 20.62 | 0.13 | 0.11 | 19.04 |
| Axis Ang | ArtGS | 0.14 | 0.04 | 0.04 | 0.02 | 1.16 | 0.01 | 0.00 | 0.20 |
| | Ours | 4.68 | 1.05 | 6.36 | 3.20 | 1.97 | 3.51 | 0.08 | 2.98 |
| | DTA | 6.51 | 2.44 | 0.31 | - | 4.20 | 0.04 | 0.04 | 2.25 |
| Axis Pos | PARIS | 0.01 | 0.00 | 0.02 | - | 0.00 | 0.00 | 0.02 | 0.01 |
| | Ours | 0.56 | 1.38 | 1.88 | - | 0.75 | 0.08 | 1.14 | 0.97 |

that our approach can accurately estimate kinematic parameters for objects with multiple degrees of freedom, thereby generalizing beyond simple articulations and proving applicable to a broader range of real-world articulated systems.

## 4.2 REAL-SCAN OBJECTS RESULTS

In the evaluation on real-world data from the two real-scan subsets (Fridge and Storage) of the PARIS dataset, which incorporate genuine scanning noise and variability not present in synthetic data, our sparse-view method (using 6 input images) exhibits competitive performance compared to state-of-the-art multi-view baselines such as ArtGS (requiring 100 views). As detailed in Table 4, across both objects, we achieve an average axis angle error of 3.93° and a position error of 0.39 cm, with significant advantages in position accuracy (outperforming ArtGS by approximately 17%) and inference time (4 minutes versus 9 minutes for ArtGS), while maintaining angular error within the same order of magnitude. This underscores the practical efficiency and robustness of our approach in constrained real-world settings.

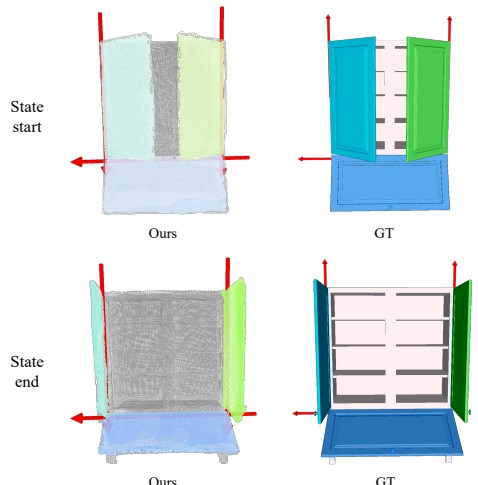

Figure 4: **Qualitative visualization on ArtGS-Multi.** We selected the Storage_45503 scene from ArtGS-Multi dataset to demonstrate our method's end-to-end capability in handling multiple rotation axes, without requiring any additional input information.

## 4.3 ABLATION STUDY

The experimental outcomes are summarized in Table 5. As a representative case, on the multi-axis object "oven_101908" (with three axes), ArtGS successfully infers all axes under the full 100-view condition but experiences a marked degradation as the number of views decreases: with 25 views per state, it correctly identifies only two of the three axes, and with 15 views per state, only one axis is accurately inferred. In general, as the number of input views drops below 15 per state, ArtGS exhibits a sharp decline in performance, largely attributable to insufficient geometric cues for reliable axis-type detection.

By contrast, our approach consistently maintains a low failure rate (below 10%) and demonstrates stable performance even with as few as 2 to 8 views on the same dataset (Table 5), highlighting its effectiveness in view-efficient reconstruction. Notably, our method can operate effectively within this range of 2-8 views; however, as the number of views decreases, the required computational time reduces accordingly, though reconstruction quality may degrade. In general, we observe that a balance between efficiency and performance is achieved around 3 views, making it a practical sweet

Table 4: **Quantitative evaluation on PARIS real-scan.** For a comprehensive comparison, we evaluate multiple methods—PARIS (Liu et al., 2023), DTA (Weng et al., 2024), ArtGS (Liu et al., 2025d), and our proposed approach—under identical experimental conditions following (Liu et al., 2025d). Lower values (↓) indicate better performance; best results are in **bold**.

| Method | Axis Ang | | | Axis Pos | | | Time (min) | | |
|--------|--------|---------|-----|--------|---------|-----|--------|---------|-----|
| | Fridge | Storage | All | Fridge | Storage | All | Fridge | Storage | All |
| PARIS | 1.90 | 30.10 | 16.00 | 0.50 | - | 0.50 | 7 | 7 | 7 |
| DTA | 2.08 | 13.64 | 7.86 | 0.59 | - | 0.59 | 29 | 29 | 29 |
| ArtGS | 2.09 | 3.47 | **2.78** | 0.47 | - | 0.47 | 9 | 9 | 9 |
| Ours | 4.54 | **3.32** | 3.93 | **0.39** | - | **0.39** | **3** | **4** | **4** |

Table 5: **Reduced Input Views Sensitivity.** The experiments were conducted on the "oven_101908" object from the ArtGS dataset, which features a total of three rotational axes. For each experiment on ArtGS and ours, images were randomly selected, and the process was repeated to derive approximate failure points as the results. Notably, the ArtGS method requires diversity in image poses; when poses are concentrated on one side, the success rate tends to decrease.

| Method | Images per state | Axes | | | Success Rate | Axis Ang | Axis Pos | Time (min) |
|--------|------------------|--------|--------|--------|--------------|----------|----------|------------|
| | | Axis 1 | Axis 2 | Axis 3 | | | | |
| ArtGS | 50/state | ✓ | ✓ | ✓ | 100% | 0.02 | 0.00 | 7 |
| | 25/state | ✓ | ✓ | × | 66.67% | - | - | 7 |
| | 15/state | ✓ | × | × | 33.33% | - | - | 7 |
| Ours | 4/state | ✓ | ✓ | ✓ | 100% | 1.65 | 0.37 | 6 |
| | 2/state | ✓ | ✓ | ✓ | 100% | 7.62 | 1.60 | 3 |
| | 1/state | ✓ | ✓ | ✓ | 100% | 13.31 | 0.82 | 2 |

spot for many real-world applications. This ablation study underscores the principal novelty of our sparse-view pipeline, wherein conventional multi-view methods exhibit significant performance degradation under the limited-view conditions typical of practical scenarios.

# 5 CONCLUSION

In conclusion, we introduce Sparse-Art, a pioneering framework that establishes a new paradigm for training-free, zero-shot reconstruction and analysis of articulated objects from sparse, un-posed images in two states. By seamlessly integrating foundation models, our approach enables unified geometric-semantic lifting, zero-shot semantic inference for part correspondence and joint classification, and lightweight classification-guided optimization for precise parameter estimation—overcoming the computational burdens and data dependencies of prior methods.

Our evaluations across synthetic and real-world benchmarks demonstrate Sparse-Art's effectiveness, with particular strengths in reliability and generalization on real-scan data. For instance, on the PARIS real-scan subset, it achieves low axis errors and reducing inference time significantly, highlighting its robustness in practical, sparse-view scenarios without domain-specific adaptations.

Despite these advances, limitations remain: the VGGT-based reconstruction could benefit from further geometric parameter optimization for enhanced accuracy in complex geometries, and overall runtime efficiency is constrained by VLM processing times, extending inference beyond ideal real-time applications. Future work may address these by incorporating faster VLM alternatives or hybrid optimization techniques, extending to multi-state dynamics or multi-object interactions.

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

## A APPENDIX

### A.1 LLM USAGE

#### A.1.1 THE USE OF LARGE LANGUAGE MODELS (LLMS)

The use of LLMs is allowed as a general-purpose assist tool. However, new this year, if LLMs played a significant role in research ideation and/or writing to the extent that they could be regarded as a contributor, then authors should describe the precise role of the LLM in a separate section on LLM usage. This section can appear in the appendix, and will not be considered as part of the page limit. Not disclosing significant LLM usage can lead to desk rejection of the paper.

Irrespective of the ways that LLMs were used in a given submission, authors should understand that they take full responsibility for the contents written under their name, including content generated by LLMs that could be construed as plagiarism or scientific misconduct (e.g., fabrication of facts). LLMs are not eligible for authorship.

#### A.1.2 OUR USAGE OF LLMS

In this paper, we primarily used a large language model (LLM) to assist with polishing the manuscript.

