# OpenReview forum: "Sparse-Art: Enabling Interactable Articulated Objects from Unposed Sparse-View Input"
_ICLR.cc/2026/Conference — ICLR 2026 Conference Withdrawn Submission_

### Official Review · Reviewer_Y2Mq · 2025-10-19

**Soundness:** 3
**Presentation:** 3
**Contribution:** 2
**Rating:** 6
**Confidence:** 4

**Summary:**

The paper introduces Sparse-Art, a novel, fully training-free and feed-forward framework for reconstructing and analyzing articulated objects from 1–4 sparse, unposed RGB images captured in two distinct articulation states. It leverages recent foundation models—specifically VGGT for 3D reconstruction, SAM for 2D segmentation, and a Vision-Language Model (VLM) for semantic reasoning—to perform unified geometric-semantic lifting, enabling zero-shot inference of part correspondences and joint types without any fine-tuning or per-object training. A lightweight, classification-guided optimizer then refines articulation parameters (e.g., joint axis and motion type) using the VLM’s predictions, significantly reducing computational cost compared to traditional optimization-heavy NeRF- or 3DGS-based methods.

**Strengths:**

The paper demonstrates that high-quality articulated object modeling can be achieved without any per-object training or fine-tuning—leveraging only off-the-shelf foundation models—making it highly accessible and scalable for real-world robotics and AR applications.
The elegant zero-shot fusion of 2D SAM masks into VGGT’s 3D point clouds via raster-order correspondence ensures semantic consistency across views and states without learnable parameters.

**Weaknesses:**

While the paper presents an interesting training-free approach to articulated object reconstruction, it has several major weaknesses that could justify rejection. First, the method relies heavily on strong assumptions—specifically, that the object is captured in exactly two distinct articulation states with 1–4 unposed images per state, which significantly limits its applicability to real-world scenarios where motion is continuous or states are unknown. The entire pipeline's performance is contingent on several large, external models (VGGT, SAM). The paper provides no ablation on the sensitivity of results to the choice or version of these models, making the core contribution feel like a brittle composition of existing tools rather than a standalone, robust method. Second, the evaluation is conducted on small-scale, mostly synthetic datasets (e.g., only 2 real-world objects in PARIS), and the comparison with baselines like ArtGS is unfairly constrained (e.g., ArtGS* is artificially downsampled to sparse views, despite being designed for dense inputs), potentially inflating the perceived advantage of Sparse-Art. Third, the core technical contributions—unified geometric-semantic lifting and VLM-guided inference—are incremental combinations of existing foundation models (VGGT, SAM, VLMs) without novel architectural or algorithmic innovations, raising questions about scientific novelty. Finally, the paper lacks ablation studies on critical components such as the impact of VLM choice, failure modes in part correspondence, or robustness to background clutter, leaving key claims about generalization and zero-shot performance inadequately substantiated.

**Questions:**

1.Input assumption limitation: The method assumes exactly two articulation states with 1–4 unposed images per state—how would Sparse-Art handle objects with more than two states, continuous motion, or unknown state boundaries? Could the framework be extended without major redesign?

2.Baseline fairness: ArtGS* was artificially constrained to sparse views, but ArtGS is designed for dense inputs—have you compared against methods natively designed for sparse views (e.g., Ditto or PARIS) under the same 6-image setting, rather than citing their dense-input results?

3.VLM dependency and robustness: The pipeline heavily relies on VLM (Gemini 2.5 Pro) for part correspondence and joint classification—how sensitive is performance to VLM choice or prompt phrasing? Can you provide failure cases where VLM misclassification leads to incorrect kinematics?

4.Real-world scalability: Only 2 real objects (Fridge, Storage) are evaluated—can you demonstrate results on a broader set of in-the-wild objects (e.g., from handheld phone captures with cluttered backgrounds) to validate generalization beyond lab-controlled scans?

5.Ablation on geometric-semantic lifting: The unified lifting via VGGT+SAM is central to your approach—can you quantify the contribution of semantic injection (e.g., by comparing against a version that uses only geometric reconstruction or naive 2D-to-3D transfer without SAM)?

---

### Official Review · Reviewer_PURC · 2025-10-28

**Soundness:** 1
**Presentation:** 2
**Contribution:** 1
**Rating:** 2
**Confidence:** 5

**Summary:**

The paper proposes a training-free pipeline for estimating the joint parameters of articulated objects from RGB images. It combines SAM to obtain part-level masks, VGGT to lift images into 3D geometric representation, and a VLM to reason about part count, cross-state correspondences, and joint type.

**Strengths:**

This work provides a training-free pipeline for articulated-object joint parameter estimation by integrating off-the-shelf modules.

**Weaknesses:**

+ This work is largely a straightforward orchestration of off-the-shelf modules with limited methodological novelty; there is no clear strategy to mitigate error propagation from upstream components (e.g., handling inaccurate SAM masks or suboptimal VGGT outputs via uncertainty modeling, visibility checks, or robust correspondence filtering).
+ The advertised advantages (training-free setup, fast inference, pose-free inputs) seem inherited from the chosen backbones rather than stemming from the proposed method itself.
+ Although prior methods report geometric reconstruction accuracy, the paper does not include direct comparisons on those metrics.
+ Relative to current SOTA, the reported results are not clearly superior and in places underperform by a noticeable margin.

**Questions:**

+ The pipeline appears to be a staged composition without gradient flow between SAM, VGGT, and the VLM. Please define precisely what you mean by “end-to-end”/“feed-forward” here.”
+ The paper lacks ablations, making it hard to assess each module’s contribution and robustness to upstream errors. Moreover, VGGT seems replaceable by classical SfM/MVS; a drop-in comparison would be informative.

---

### Official Review · Reviewer_GwrY · 2025-11-01

**Soundness:** 2
**Presentation:** 2
**Contribution:** 2
**Rating:** 4
**Confidence:** 2

**Summary:**

The paper introduces Sparse-Art, a fully training-free pipeline for articulated object reconstruction and analysis from 1–4 unposed RGB views per state (two states per object). It lifts 2D semantics from SAM into VGGT pointmaps to build a semantically enriched 3D representation, uses a VLM to infer part counts/correspondences and joint types in zero-shot mode, and estimates articulation with a lightweight, classification-guided optimizer including a distance-proportional weighting for revolute joints. Experiments on PARIS, DTA-Multi, and ArtGS-Multi show competitive performance under sparse views and minutes-level runtime on real scans.

**Strengths:**

This work proposes a training-free method for articulated object reconstruction, which is flexible and practical. Moreover, it novelly involves VLM for articulated object reconstruction and demonstrates its effectiveness, which may provide some insights for future work.

**Weaknesses:**

The experimental results are not convincing. As shown in Table 2, they fall far behind ArtGS in part segmentation performance, especially for multi-part objects. There is a lack of novelty, as the work heavily depends on off-the-shelf methods.

**Questions:**

1. Please provide a runtime breakdown per stage (VGGT, SAM, VLM, optimizer) and the hardware usage.
2. Do you plan to release the code for better reproducibility?
3. How stable are the results across different VLMs (e.g., open-source options)? Please provide a small ablation study with latency and cost.

---

### Official Review · Reviewer_kqKq · 2025-11-09

**Soundness:** 3
**Presentation:** 3
**Contribution:** 3
**Rating:** 6
**Confidence:** 4

**Summary:**

The paper proposed a zero shot sparse-view articulated object reconstruction as well as movable part segmentation and part motion estimation. The approach utilizes a setting recently introduces which relies on initially captured multi-views of starting and ending states of articulated objects. It relies on zero-shot VLMs, SAM and 3D foundation model i.e. VGGT for holistic part-level 3D point cloud reconstruction and motion estimation. Experiments are conducted and results demonstrated on both single and multi-articulated objects in both synthetic and real domain.

**Strengths:**

In my opinion, the paper has the following strengths:

1. A good application of existing foundation models for zero-shot method holistic part-level 3D reconstruction without training or finetuning.

2. Significant improvement in most of the quantitative metrics and qualitative results over competing baselines for articulated motion estimation.

3. Extensive experimental results are shown on both single and multi-articulated objects as well synthetic and real domains.

4. The paper is nicely written, easy to follow and the figures complement the text nicely.

**Weaknesses:**

In my opinion, the paper has the following weaknesses:

1. The paper reconstructs pointclouds but doesn't report chamfer distance metrics compared to competing baselines. Is that a limitation of the method and that should be clearly stated since other approaches can reconstruct articulated object meshes which are highly useful for instance for real2sim efforts downstream for instance for integrating them in downstream physics simulators or re-rendering RGB from multiple viewpoints.

2. Since the pipeline is highly modular and zero-shot, there is a fair chance the errors accumulate. For instance the VLM may not be generalizable for certain type of articulated objects etc. I did not see a clear discussion on failure modes as well as how the authors plan to address error accumulation from various modular methods.

3. Why is the DTA multi fridge result i.e. for real scene in Table 3 worse than baselines which looks like it is an outlier result compared to the rest of trend on synthetic scenes.

4. Can the authors comment on the runtime analysis of their method and how does it compare to competing baselines especially with the feed-forward approaches [1] for a fair comparison i.e. if there is an efficiency vs accuracy trade-off here.

5. Does the VLM need specific prompt tuning to get accurate results?

6. The paper mentions keypoint correspondances. Does it fail when parts are heavily occluded considering the authors are only considering 1-4 view during inference? and how are those failure modes handled?

[1] Heppert et al. CARTO: Category and Joint Agnostic Reconstruction of ARTiculated Objects, CVPR 2023

**Questions:**

Please see all questions in the weakness section. I look forward to seeing author's responses in the rebuttal.

---

### Note · Authors · 2025-11-14

I have read and agree with the venue's withdrawal policy on behalf of myself and my co-authors.